# Perturbations of Lipids and Oxidized Phospholipids in Lipoproteins of Patients with Postmenopausal Osteoporosis Evaluated by Asymmetrical Flow Field-Flow Fractionation and Nanoflow UHPLC–ESI–MS/MS

**DOI:** 10.3390/antiox9010046

**Published:** 2020-01-05

**Authors:** Kang Geun Lee, Gwang Bin Lee, Joon Seon Yang, Myeong Hee Moon

**Affiliations:** Department of Chemistry, Yonsei University, Seoul 03722, Korea; eagle52109@yonsei.ac.kr (K.G.L.); yigb1@yonsei.ac.kr (G.B.L.); js0610@yonsei.ac.kr (J.S.Y.)

**Keywords:** oxidized lipid, lipoprotein, postmenopausal osteoporosis, asymmetrical flow field-flow fractionation, nUHPLC–ESI–MS/MS

## Abstract

Osteoporosis, a degenerative bone disease characterized by reduced bone mass and high risk of fragility, is associated with the alteration of circulating lipids, especially oxidized phospholipids (Ox-PLs). This study evaluated the lipidomic changes in lipoproteins of patients with postmenopausal osteoporosis (PMOp) vs. postmenopausal healthy controls. High-density lipoproteins (HDL) and low-density lipoproteins (LDL) from plasma samples were size-sorted by asymmetrical flow field-flow fractionation (AF4). Lipids from each lipoprotein were analyzed by nanoflow ultrahigh performance liquid chromatography–electrospray ionization–tandem mass spectrometry (nUHPLC–ESI–MS/MS). A significant difference was observed in a subset of lipids, most of which were increased in patients with PMOp, when compared to control. Phosphatidylethanolamine plasmalogen, which plays an antioxidative role, was increased in both lipoproteins (P-16:0/20:4, P-18:0/20:4, and P-18:1/20:4) lysophosphatidic acid 16:0, and six phosphatidylcholines were largely increased in HDL, but triacylglycerols (50:4 and 54:6) and overall ceramide levels were significantly increased only in LDL of patients with PMOp. Further investigation of 33 Ox-PLs showed significant lipid oxidation in PLs with highly unsaturated acyl chains, which were decreased in LDL of patients with PMOp. The present study demonstrated that AF4 with nUHPLC–ESI–MS/MS can be utilized to systematically profile Ox-PLs in the LDL of patients with PMOp.

## 1. Introduction

Osteoporosis, a chronic degenerative disease, is characterized by reduced bone mass, which leads to an increased risk of bone fracture [1,2,3]. Osteoporosis is recognized as a major public health problem worldwide, as it limits patient mobility to a greater degree than strokes and myocardial infarction [2]. Over their lifetime, about 30% of women and 12% of men are expected to develop osteoporosis [3]. This, combined with the fact that the elderly population is increasing, places a major economic burden on society. Several factors are known to cause osteoporosis, including hormonal changes, lack of physical exercise, drug use, and an inadequate supply of nutrients that are essential for bone formation [2]. Estrogen deficiency is a major risk factor in menopausal women, since the decrease in estrogen levels perturbs the activity of bone remodeling, eventually leading to low bone density [4,5]. Bone loss is associated with increased bone resorption of osteoclasts and decreased bone formation of osteoblasts [6]. Additionally, the activity of osteoclasts is enhanced by oxidative stress from reactive oxygen species (ROS) [7], and antioxidants inhibit the effects of minimally oxidized low-density lipoproteins (MM-LDL) [8,9]. Since the oxidation of lipoproteins occurs at lipids, bone loss should be associated with alterations in circulating lipid profiles, especially oxidized lipids. Indeed, a correlation between lipid changes and postmenopausal osteoporosis (PMOp) has been found in the femur tissue of osteoporotic mice [10] and human plasma samples [11,12]. Recently, it has been found that inflammatory bioactive lipids, which promote atherosclerosis, also induce bone loss [13], and that oxidized LDL is pathogenically associated with the loss of bone mass [14]. However, there has not yet been a systematic evaluation of oxidized lipids in relation to PMOp based on lipidomic changes at the lipoprotein level.

Lipoproteins, globular complexes carrying lipids and a few proteins, circulating in the blood, are classified as either high-density lipoproteins (HDL), LDL, or very low-density lipoproteins (VLDL). Several lipid classes, such as phosphatidylcholine (PC), lyso-PC (LPC), ceramide (Cer), and cholesteryl ester (CE), are known to be risk factors for cardiovascular diseases [15,16,17]. Oxidation of LDL by ROS has been recognized to promote atherosclerotic plaque formation in the arterial wall [18,19]. Since the oxidation of LDL is correlated with osteoporosis, it is important to examine lipid changes, as well as lipid oxidation in circulating lipoproteins.

In this pilot study, lipid profiles at the lipoprotein level were quantitatively analyzed, to examine lipoprotein-dependent lipid alterations and the degree of lipid oxidation in the plasma of patients with PMOp. HDL and LDL particles from the plasma of patients with PMOp were size-sorted by semi-preparative scale asymmetrical flow field-flow fractionation (asymmetrical FlFFF or AF4), an elution-based method which separates macromolecules or particulate species by size [20,21,22,23,24]. Then, collected fractions of narrow size distribution can be used for proteomic or lipidomic analysis, as reported by the previous studies [24,25]. Lipids in the collected lipoprotein fractions in this study were analyzed by nontargeted identification, followed by targeted quantification with nanoflow ultrahigh performance liquid chromatography–electrospray ionization–tandem mass spectrometry (nUHPLC–ESI–MS/MS). Lipid molecules whose levels were significantly different between patients and healthy controls were determined to be the lipoprotein-specific lipids that are relevant to PMOp. In addition, lipid oxidation was systematically investigated by quantitatively analyzing the possible oxidation products of phospholipids, which were significantly decreased in the LDL of patients with PMOp.

## 2. Materials and Methods

### 2.1. Chemicals 

Standard materials of bovine serum albumin (BSA), HDL, and LDL were purchased from Sigma-Aldrich Co. (St. Louis, MO, USA). HPLC grade solvents used for nUHPLC–ESI–MS/MS and lipid extraction were purchased from J.T. Baker, Inc. (Phillipsburg, NJ, USA): CH_3_OH, H_2_O, 2-propanol (IPA), CHCl_3_, CH_3_CN, HCOOH, NH_4_OH, NH_4_HCO_3_, NH_4_HCO_2_, and methyl-*tert*-butyl ether (MTBE). 

A total of 36 lipid standards purchased from Matreya, LLC. (Pleasant Gap, PA, USA) and Avanti Polar Lipids, Inc (Alabaster, AL, USA) were utilized for the optimization of nUHPLC–ESI–MS/MS run conditions: lysophosphatidylcholine (LPC) 16:0, LPC 17:0, phosphatidylcholine (PC) 13:0/13:0, PC 16:0/16:0, PC 18:1/18:0, lysophophatidylethanolamine (LPE) 18:0, LPE 17:1, phosphatidylethanolamine (PE) 16:0/16:0, PE 17:0/17:0, PE plasmalogen (PEp) P-18:0/22:6, lysophosphatidic acid (LPA) 14:0, LPA 17:0, phosphatidic acid (PA) 12:0/12:0, PA 17:0/17:0, lysophosphatidylglycerol (LPG) 14:0, LPG 17:1, phosphatidylglycerol (PG) 15:0/15:0, PG 16:0/16:0, lysophosphatidylinositol (LPI) 20:4, phosphatidylinositol (PI) 16:0/18:2, PI 12:0/13:0, sulfohexosylceramide (SulfoHexCer) d18:1/12:0, SulfoHexCer d18:1/17:0, ceramide (Cer) d18:1/14:0, Cer d18:1/17:0, monohexosylceramide (HexCer) d18:1/16:0, HexCer d18:1/17:0, dihexosylceramide (Hex2Cer) d18:1/16:0, sphingomyeline (SM) d18:1/16:0, SM d18:1/17:0, cardiolipin (CL) (14:0)_4_, CL (14:1)_3_(15:1), diacylglycerol (DG) 16:0/18:1, DG 17:0/17:0-D_5_, triacylglycerol (TG) 18:0/18:0/18:1, and TG 17:0/17:1/17:0-D_5_. Standard lipids with odd-numbered fatty acyl chain were utilized as internal standards added to lipid extracts for quantification. Fused silica capillaries with an inner diameter of 20, 50, 100, and 200 μm (all had identical outer diameters of 360 μm) were purchased from Polymicro Technology (Phoenix, AZ, USA).

### 2.2. Plasma Samples

Female plasma samples from patients with PMOp but no other medical history (age = 60.6 ± 7.9; BMI = 22.7 ± 2.3; *n* = 10) and healthy postmenopausal controls (age = 72.1 ± 3.9; BMI = 24.8 ± 1.7; *n* = 10) were collected with informed consent from Chungbuk National University Hospital (Cheongju, Korea) and Chungnam National University Hospital (Daejeon, Korea), respectively, and provided by Biobank of Korea. Participants were free from the use of drugs, such as bisphosphonates or statins. The study was conducted in accordance with the Declaration of Helsinki, and the protocol was approved by the Yonsei University Institutional Review Board (7001988-201805-BR-371-01E, 9 May 2018). Detailed demographic data are in Appendix A. All plasma samples were kept at −80 °C. Before fractionating lipoproteins by AF4, each plasma sample was treated with a ProteoPrep^®^ Immunoaffinity Albumin & IgG Depletion Kit from Sigma-Aldrich to eliminate IgG and albumin. For the UV detection of HDL and LDL in AF4 separation only, the depleted plasma samples were stained with Sudan Black B (SBB) solution. For the collection of each lipoprotein fraction, the depleted plasma samples without staining were directly injected into AF4.

### 2.3. Separation of HDL and LDL by AF4

Plasma lipoproteins (HDL and LDL) were separated by a semi-preparative scale AF4 channels from Wyatt Technology Europe GmbH (Dernbach, Germany) with channel dimensions of 26.6 cm (length) × 250 µm (thickness) with a trapezoidal decrease in channel width from 4.0 to 1.0 cm. A regenerated cellulose membrane with a nominal molecular weight cut-off of 10 kDa from Millipore (Danvers, MA, USA) was layered above the stainless frit at the accumulation wall. The carrier solution of AF4 was 0.1 M of PBS prepared from deionized water (>18 MΩ·cm) and filtered with a 0.22 μm nitrocellulose membrane from EMD Millipore (Billerica, MA, USA) prior to use. Sample injection was made with a loop injector (model 7125, Rheodyne, Cotati, CA, USA), which was connected between an HPLC pump (model SP930D, Young-Lin Instrument, Seoul, Korea) and the AF4 channel. For sample relaxation, the focusing/relaxation procedure was applied by conducting the flow from the inlet at a 1:9 ratio respective to the outlet of the channel. The sum of the two flow rates was adjusted to be the same as the crossflow rate used for AF4 separation of lipoproteins. Rates of outflow/crossflow were 0.4/3.6 in mL/min. Detection of stained lipoproteins during AF4 separation was made at a wavelength of 600 nm, using a UV-Vis detector (model YL9120, Young-Lin Instrument). Data were recorded by Autochro-Win 3.0 plus (Young-Lin Instrument).

### 2.4. Lipid Extraction

Each lipoprotein fraction was concentrated to about 500 μL at 2000× *g* for 10 min, using Amicon Centrifugal Filters (MWCO 10 kDa; Millipore, Burlington, MA, USA). Extraction of lipids from the HDL and LDL fractions collected from AF4 followed the two-stage extraction method, using MTBE/CH_3_OH [25,26]. After lipid extraction, lipids in organic solvents were transferred to a 2 mL tube and sealed with 0.45 μm MilliWrap PTFE membrane from Millipore, to avoid lipid evaporation during lyophilization for 12 h. Then, dried lipid powders (1.0~1.2 mg) were reconstituted in CH_3_OH:CHCl_3_ (9:1, v/v), to prepare a stock solution for each fraction, which was stored in a −80 °C freezer. The stock solution was diluted with CH_3_OH:H_2_O (8:2, v/v), at a final lipid concentration of 5 μg/μL, prior to analysis.

### 2.5. Nanoflow UHPLC–ESI–MS/MS

Lipid analysis was carried out by two-stage nUHPLC–ESI–MS/MS analysis as follows: nontargeted identification of lipid species contained in lipoprotein fractions from each pooled plasma sample was performed by using a Dionex Ultimate 3000RSLCnano System with an LTQ Velos ion trap MS (Thermo Scientific, San Jose, CA, USA), followed by targeted quantification of individual samples based on the selected reaction monitoring (SRM) method, using a model nanoACQUITY UPLC system from Waters (Milford, MA) with a TSQ Vantage triple-stage quadrupole MS from Thermo Scientific. Both nUHPLC–ESI–MS/MS analyses utilized an identical capillary LC column, a homemade 7 cm long pulled-tip capillary column (100 μm inner diameter) packed in the laboratory with 1.7 μm ethylene bridged hybrid (BEH) C18 particles (130 Å), which were unpacked from a Waters^TM^ ACQUITY UPLC^®^ BEH C18 column. Prior to packing, a 0.5 cm portion of the column tip was filled with 3 μm 100 Å Watchers^®^ ODS-P C18 particles (Isu Industry Corp, Seoul, Korea) as a self-assembled frit.

The mobile phase solutions used for binary gradient elution were H_2_O:ACN (9:1, v/v) for A and IPA:CH_3_OH:ACN (6:2:2, v/v/v) for B, both of which were added with a mixture of ionization modifiers (5 mM of ammonium formate and 0.05% ammonium hydroxide), which can be utilized in both positive- and negative-ion modes. Sample loading was made to the analytical column directly by using 99% mobile phase A at a flow rate of 700 nL/min for the first 10 min. For nontargeted analysis, 1 μL of lipid extracts (5 μg/μL) spiked with a mixture of internal standards (500 fmol each) was injected. After sample loading, the pump flow rate was increased to 7 μL/min, with the split valve on, so that the column flow rate was set to 300 nL/min. In the positive-ion mode, gradient elution began by ramping up mobile phase B to 70% for 2 min, and then mobile phase B was linearly increased to 80% for 3 min, 85% for 5 min, 90% for 10 min, and maintained at 99% for 13 min. In the negative-ion mode, mobile phase B was ramped to 75% for 2 min, 80% for 2 min, 85% for 4 min, 90% for 10 min, and was maintained at 99% for 10 min. The m/z range of a precursor MS scan was set to 300–1000 amu, with an ESI voltage of 3.0 kV for both the positive- and negative-ion modes. Data-dependent collision-induced dissociation (CID) experiments were performed at 40% normalized collision energy. The determination of lipid molecular structure was based on CID spectra, using LiPilot, a PC-based software developed in our laboratory [27], and confirmed manually.

For targeted quantification, each lipid extract sample of an individual human sample spiked with 16 different internal standards (2 pmol/μL) was analyzed by selected reaction monitoring (SRM). After sample loading at 700 nL/min mobile phase A for 10 min, the pump flow rate was increased to 15 μL/min, to reduce dwell time. The column flow was adjusted to 300 nL/min as before, using the split valve. Since SRM quantifications were made in the polarity switching mode, which detects ions at the positive- and negative-ion mode alternatively, an identical gradient elution method was used: mobile phase B was ramped to 70% for 6 min, further to 80% for 9 min, and was maintained at 99% for 15 min; and then the mobile phase composition was returned to 100% mobile phase A, and the column was reconditioned for 5 min. Lipid classes of LPC, PC, LPE, PE, SulfoHexCer, SM, Cer, HexCer, DG, and TG were detected in the positive-ion cycle of the polarity switching mode, and those of LPA, PA, LPG, PG, LPI, PI, and CL in the negative-ion cycle. Collision energy was assigned differently as 20~45 V, depending on the lipid classes listed in Appendix A. Statistical analysis was carried out by using SPSS software (version 24.0, IBM Corp., Armonk, NY, USA) for Student’s *t*-test and Minitab 17 software for principal component analysis (PCA).

## 3. Results and Discussion

### 3.1. Size Separation of Lipoproteins by Semi-Prep AF4

Plasma lipoproteins were separated by semi-prep scale AF4 (Figure 1), indicating that plasma HDL particles were clearly separated from LDL for both patients with PMOp and controls. The LDL peaks in both groups appeared mostly as single peaks with slight tailing, suggesting an incomplete separation of VLDL particles. However, the peak intensities of LDL from patients with PMOp were much higher than those of controls. In addition, the average retention time of LDL peaks was 17.9 ± 0.5 min (*n* = 10) for patients with PMOp, which was very close to the 18.1 ± 0.2 min (*n* = 10) for controls, suggesting that the size of LDLs was not significantly reduced upon the development of PMOp. This is clearly different from the typical size reduction of LDL particles in patients with coronary artery diseases [28,29]. However, the concentration of LDL was increased in patients with PMOp about 2.2-fold compared to that of controls based on the peak area comparison. Each lipoprotein fraction was collected during the two consecutive AF4 runs for lipid analysis with nUHPLC–ESI–MS/MS.

### 3.2. SRM-Based Quantification of Lipids in Each Lipoprotein

Lipid extracts of each lipoprotein fraction were analyzed for nontargeted identification of lipids by nUHPLC–ESI–MS/MS. The base peak chromatograms of lipid standards and lipid extracts of HDL and LDL fractions are shown in Appendix A, respectively. A total of 379 lipid molecular structures were identified from MS/MS spectra obtained from data-dependent collision-induced dissociation experiments during nUHPLC–ESI–MS/MS. Among them, 252 lipid species were quantified from each individual sample with SRM-based nUHPLC–ESI–MS/MS runs. The precursor ions and product ions of each lipid class used for SRM quantification are listed in Appendix A. Briefly, the limit of detection (LOD)/limit of quantitation (LOQ) values of lipid classes range from 0.005/0.015 pmol (LPC 17:0) to 0.041/0.135 pmol (CL (14:0)_4_) as listed in Appendix A. Lipid species calculated to be below the LOQ of each lipid class were removed from the quantified data lists. The quantified amount of each lipid is reported in Appendix A as the corrected peak area, the ratio of the peak area of an individual lipid species in comparison to that of an internal standard (IS) for each class, which is a good estimate of the pmol amount of each species. PC, PE, and TG were quantified by the total length of their acyl chains with double bond numbers and the detailed isomeric acyl chain structures identified from the data-dependent CID experiments are listed in Appendix A.

### 3.3. Lipid Alterations in Each Lipoprotein from Patients with PMOp

The difference in lipid amounts between patients with PMOp and controls was examined with PCA by comparing the overall lipid profiles in HDL and LDL based on lipid species (Figure 2). Perturbation in plasma lipid profiles of patients with PMOp was larger in HDL than in LDL, and the differences among individuals were larger in patients with PMOp than in controls. Variations in individual lipid levels demonstrated that changes in the accumulation of lipids were largely made upon the development of PMOp (Figure 3).

The overall amount of PC and LPA (Figure 4a) showed more than a two-fold increase in the HDL of patients with PMOp, while PEp (Figure 4b) was increased in both HDL and LDL. The four lipid classes (Cer, Hex2Cer, DG, and TG) (Figure 4c) were largely increased in LDL only. High-abundance lipids were defined by a relative abundance > 100%/(total number of lipids within the class). The 12 remaining lipid classes were not significantly different between patients with PMOp and controls (Appendix A). Since the overall amounts of PC and DG were an order of magnitude greater than other lipid classes, their increase in patients with PMOp may be a major contributor to lipid accumulation in HDL and LDL, respectively. Individual lipid species showing more than a 1.5-fold change with *p* < 0.05 in the PMOp group, either in the HDL or LDL fraction, are plotted in a heatmap representing individual variations (Appendix A).

Lipid species that varied significantly (>1.5-fold with *p* < 0.05) in the PMOp group either in the HDL or LDL fraction are listed in Appendix A. The six PC species (34:1, 34:2, 36:2, 36:3, 38:4, and 40:6), LPI 16:0, and SM d18:1/22:0 were significantly increased in HDL, and PI 18:0/20:4 was decreased in HDL (Figure 5).

The three PEp species (P-16:0/20:4, P-18:0/20:4, and P-18:1/20:4) in this study increased more than two-fold in both lipoprotein fractions (*p* < 0.05). By comparing the relative amounts of these three PEp species (Figure 4), as well as their relative abundance (Appendix A), it was found that each of the three species was present in similar amounts, and together they composed nearly 92% of total PEp. PEp is known to protect the unsaturated acyl chain of lipids against oxidative damage from ROS [30]. The increase in PEp was similarly reported in menopausal women with low bone-mineral density (BMD) compared to the BMD in women with normal bone density in an earlier study [12]. While the last study showed a significant increase of PE P-18:0/20:4 species, the present work showed more detailed information on the accumulated PEp species.

LPA 16:0, PA 18:1/22:6, and HexCer d18:1/16:1 were increased in HDL but significantly decreased in LDL. LPA 16:0 was the most abundant LPA species identified, with a 2.3-fold increase in HDL, but a 2.4-fold decrease in LDL. The total amount of LPA in HDL was also elevated (~3-fold). However, the total amount of LPA in LDL was not significantly decreased (Figure 4), due to an increase in LPA 18:0. This results in an increase in the overall LPA in the plasma of patients with PMOp. LPA has been reported to play an important role in the pathophysiology of osteoporosis by promoting the integration of osteoclasts [31]. An increase of LPA 16:0 in our study corroborated the results of a recent study that showed an increase of LPA 16:0 in the plasma from an age-induced osteoporosis mouse model [32].

Cer d18:1/16:1 appeared to be significantly decreased in both lipoprotein fractions (Figure 5). The relative abundance of Cer d18:1/16:1 was 21% (Appendix A) in HDL and 2.7% in LDL based on the control group. However, compared to the other three high-abundance Cer species (d18:1/22:0, d18:1/24:0, and d18:1/24:1; Figure 4), a decrease of Cer d18:1/16:1 in LDL was negligible to the total Cer level in LDL, and the three most abundant species were further increased. Although these three Cer species contributed to the significant (~ 2-fold with *p* < 0.05) increase in the overall Cer level in LDL, they were not significantly different when assessed as individual lipid species (*p*-values as 0.06, 0.064, and 0.113, respectively). Cer is involved with the regulation of cell differentiation, proliferation, and apoptosis. A recent report on the aging-related lipidomic changes in human bone mesenchymal stem cells (BMSCs) showed that PC, PE, and Cer levels were significantly elevated with aging [33]. Cer levels were reported to be increased in the femur of a PMOp mouse model [10] and in the serum (Cer d18:0/18:0) of an ovariectomy-induced osteoporosis rat model [34]. These results are in line with the observed increase in most Cer species in patients with PMOp.

The two TG (50:4 and 54:6) species were increased in LDL only. Although the total TG amounts were not changed in HDL in our study, they were significantly increased in the LDL fraction due to contributions from two highly unsaturated TG species (50:4 and 54:6). While the relationship between the increase of TG and the development of PMOp is not clearly understood, similarly to our study, the overall TG level has been reported to increase in postmenopausal women [35]. Moreover, TG species with more unsaturated acyl chains appear to be more influenced than those with saturated chains. Typically, an increased level of total blood plasma TG is associated with an increased risk of metabolic diseases, such as cardiovascular disease and type 2 diabetes. However, the increase of TG was mostly attributable to TG with more saturated acyl chains [36,37]. As such, although an increase in TG with more unsaturated acyl chains in LDL may be a characteristic difference in patients with PMOp, the biological mechanism and repercussions of this increase are not yet clearly understood.

While HexCer d18:1/16:1 was increased in HDL and decreased in LDL, three other HexCer species (d18:1/22:0, d18:1/24:0, and d18:1/24:1) showed the exact opposite pattern. HexCer belongs to a group of glycosphingolipids which is considered to be essential in the formation of osteoblasts [38], and an increased level of glycosphingolipids can accelerate osteoblast differentiation, which may disturb the balance between bone resorption and formation in bone tissue [10]. Although the four HexCer species in this study showed different trends in both lipoproteins, the change in the total amount of HexCer in each lipoprotein fraction was negligible (Appendix A) due to compositional alterations. Since the roles of individual HexCer species have not been examined elsewhere, the relationship between HexCer and PMOp is not clear.

While most species were increased in lipoprotein fractions, there were a few species that were decreased in the LDL fraction, namely PC 38:5 and PA 18:1/22:6 (Figure 5). These were further examined for lipid oxidation.

### 3.4. Quantification of Ox-PLs from the LDL of Patients with PMOp

In order to further investigate the degree of lipid oxidation upon the development of PMOp, lipid species that were significantly decreased in LDL were examined. Among these, those which contained unsaturated (more than two double bonds) acyl chain in the sn-2 position were selected as candidate species for lipid oxidation, since lipid oxidation occurs primarily at unsaturated acyl chains. The selected PL species were PC (34:5 and 38:5), PA (16:0/18:2 and 18:1/22:6), and PI (18:0/18:2 and 18:0/22:4). The Ox-PL products originating from each intact PL species were determined from the original CID spectra, including Ox-PL species produced by hydroxylation, hydroperoxylation, and truncation of acyl chains followed by chain termination with either aldehyde or carboxylic acid form.

The molecular structures of Ox-PC species generated from an intact PC 16:0/22:5 can be identified from their characteristic fragment ions (Figure 6). In general, intact PC 16:0/22:5 (*m/z* 808.5) molecules under CID experiment resulted in fragment ions (Figure 6a) after the loss of the phosphocholine head group at m/z 570.3 ([M+H–183]^+^), the dissociation of the acyl chain in the form of carboxylic acid ([M+H–RCOOH]^+^) either from the sn-1 (*m/z* 552.3) or sn-2 position (*m/z* 478.3), and in the form of ketene ([M+H–R’CH=C=O]^+^) at m/z 570.3 and 496.3, respectively. The CID spectrum of the singly hydroxylated form (m/z 824.5) (Figure 6b) of the same PC molecule shows a base peak that indicates the loss of water ([M+H–H_2_O]^+^) at m/z 806.3, along with the loss of the sn-1 acyl chain in the form of carboxylic acid at m/z 568.3 and in the form of ketene at m/z 586.3, supporting the addition of one oxygen as PC 16:0/22:5+O (+O represents the addition of a hydroxide to one of the double bonds in the 22:5 acyl chain). The CID spectrum of the hydroperoxylated product of the same PC molecule shows similar fragment ion patterns as observed in the hydroxylated species, plus an additional fragment ion at *m/z* 808.3, which shows the loss of O_2_ ([M+H–O_2_]^+^) from the parent ion (Figure 6c), supporting the structure of PC 16:0/22:5+OO (+OO for hydroperoxylation). A short chain product after the truncation of the same intact PC molecule (Figure 6d) possesses a characteristic fragment ion produced after the loss of CO_2_ from the parent ion at *m/z* 754.3 ([M+H–CO_2_]^+^), along with a similar fragment ion pattern after the loss of an acyl chain, supporting the formation of terminal carboxylic acid at the sn-2 acyl chain after the truncation of the original acyl chain (22:5) into 19:4 as PC 16:0/19:4COOH.

A total of 20 Ox-PC, 6 Ox-PA, and 7 Ox-PI species were selected for targeted quantification analysis in separate nUHPLC–ESI–MS/MS runs based on SRM. Quantified results of the 33 Ox-PL species are listed in Table 1, along with the corrected peak area and the P/C ratio. Among them, 12 Ox-PL species were found to be significantly (>1.5-fold with *p* < 0.05) increased in patients with PMOp compared to controls and originated from the six abovementioned relatively abundant PL species. The nine Ox-PC species appeared to originate from four different isomeric structures of PC 38:5 (16:0/22:5, 18:0/20:5, 18:1/20:4, and 18:2/20:3), the two Ox-PA species originated from PA 18:1/22:6, and an Ox-PI was from PI 18:0/18:2, all of which displayed a relatively abundant original intact species. These results indicate that significant lipid oxidation occurred in PLs with highly unsaturated acyl chains, which were decreased in the LDL of patients with PMOp when compared to controls. While earlier studies reported that Ox-LDL and the increase of lipoxygenases (a family of enzymes generating oxidized polyunsaturated fatty acids with age) stimulated apoptosis of osteoblastic cells [39,40,41], the present work revealed that increasing Ox-PLs in the LDL of patients may be associated with the development of PMOp.

## 4. Conclusions

This study demonstrated that a combination of FlFFF and nUHPLC–ESI–MS/MS can be utilized for the size-sorting of lipoproteins in blood plasma samples and for comprehensive analysis of lipoprotein-dependent lipid alteration. FlFFF analysis of lipoproteins showed that LDL sizes were not significantly reduced in patients with PMOp compared to controls, based on the small differences in retention times. However, LDL levels increased by more than two-fold. SRM-based quantification of a total of 252 lipids from 379 identified species revealed that, among lipids that differed between patients with PMOp and control, most were increased in patients with PMOp. The total amount of each lipid class increased significantly (>1.5-fold with *p* < 0.05) in patients with PMOp, especially in the classes of PC and LPA in HDL, PEp in both lipoprotein fractions, and Cer, Hex2Cer, DG, and TG in LDL. At the molecular level, 22 lipid molecules were significantly altered in the levels, and most were increased in either HDL, LDL, or both. Only PI 18:0/20:4 and the three HexCer (d18:1/22:0, d18:1/24:0, and d18:1/24:1) were decreased in HDL, with the three HexCer showing an increase in LDL. Of particular interest is PEp, which is known to play an antioxidative role against lipid oxidation. Similar to an earlier study showing an increase of PE (P-18:0/20:4) in menopausal women with low BMD [12], our study showed significant increases in PE (P-16:0/20:4, P-18:0/20:4, and P-18:1/20:4) in both lipoprotein fractions of patients with PMOp. Therefore, the elevation of PEp levels strongly supports the hypothesis that lipid oxidation is increased in patients with PMOp. Further investigation into the Ox-PL products in the LDL fraction from patients with PMOp showed that significant increases in lipid oxidation were observed in PLs with unsaturated acyl chains, which were significantly reduced in the LDL fraction from patients with PMOp. Among 33 Ox-PL species examined, the 12 Ox-PL molecules produced from the abundant PC 38:5, PA 18:1/22:6, and PI 18:0/18:2 species were increased more than 1.5-fold (*p* < 0.05) when compared to postmenopausal healthy controls.

The present study not only showed the capability of analyzing lipoprotein-dependent analysis of plasma lipids by coupling AF4 with nUHPLC–ESI–MS/MS, but also demonstrated that oxidized lipid products can be monitored at the molecular level. These could be used in the future as specific indicators to distinguish the developmental stage of PMOp. The present study utilized a limited number of patients for this pilot study; a further systematic investigation, with a large cohort, will be needed to validate the usefulness of the candidate lipids, including Ox-PLs. For high-speed analysis of osteoporosity-specific lipids, including Ox-PLs, it would be desirable to implement a top-down lipid analysis method with a miniaturized AF4 system directly hyphenated to ESI–MS/MS, so that size-sorted lipoproteins from AF4 can be directly fed to MS for lipid-target-specific analysis. This will require a comprehensive investigation of MS efficiency for significantly altered lipid species, such as Ox-PLs.

## Figures and Tables

**Figure 1 antioxidants-09-00046-f001:**
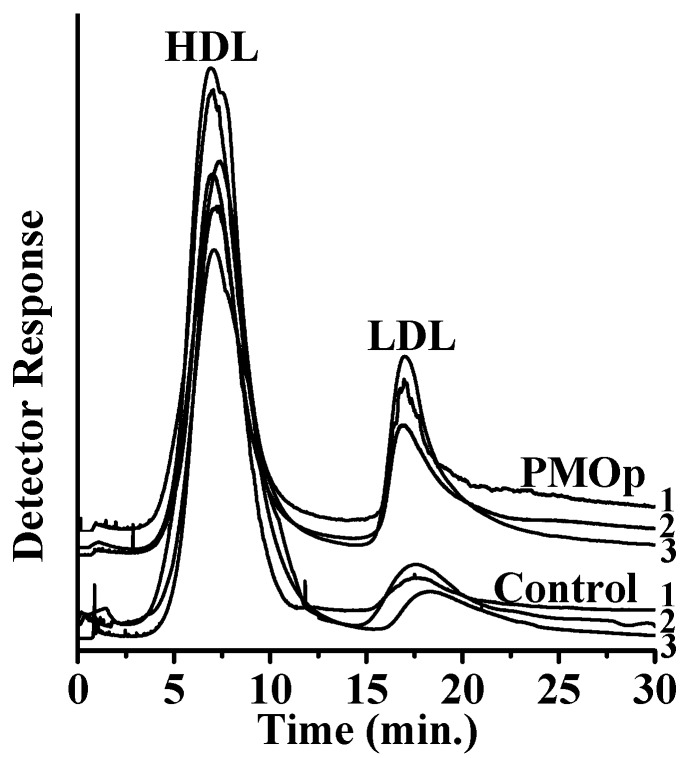
Fractograms of plasma samples (stained with Sudan Black B) from patients with PMOp and controls of postmenopausal females showing the separation of HDL and LDL by semi-prep AF4 obtained at outflow rate/crossflow rate = 0.4/3.6 in mL/min. Detections were at λ = 600 nm.

**Figure 2 antioxidants-09-00046-f002:**
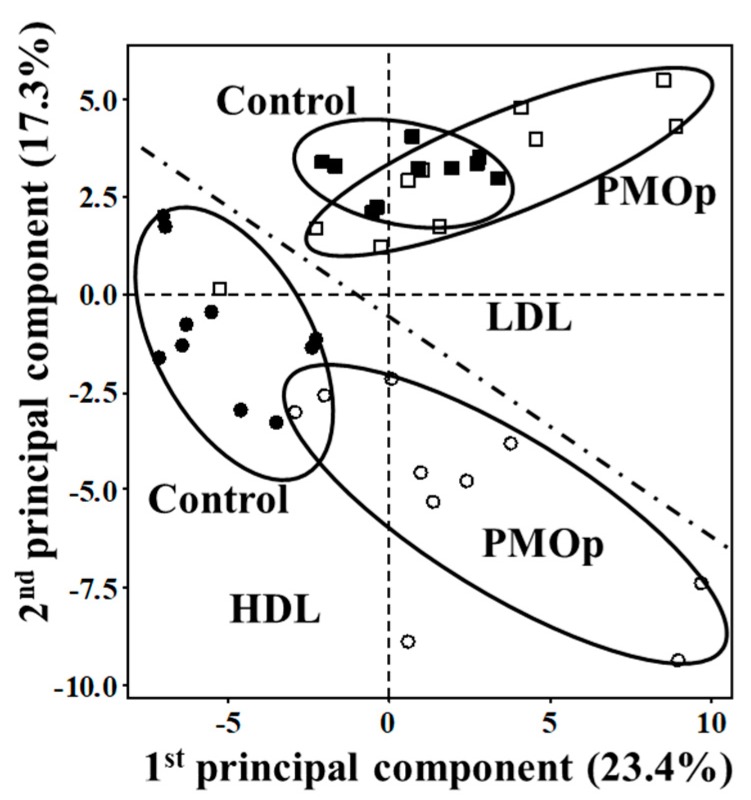
PCA plots representing the differences in lipid profiles of HDL and LDL fractions between patients with PMOp (open symbols) and control (filled symbols).

**Figure 3 antioxidants-09-00046-f003:**
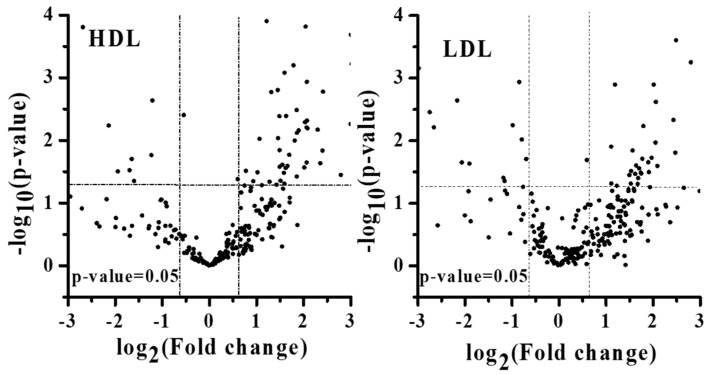
Volcano plots [-log_10_(*p*-value) vs. log_2_(fold change)] of quantified lipids (**a**) HDL and (**b**) LDL fractions. Fold change represents the ratio of the patient group to control.

**Figure 4 antioxidants-09-00046-f004:**
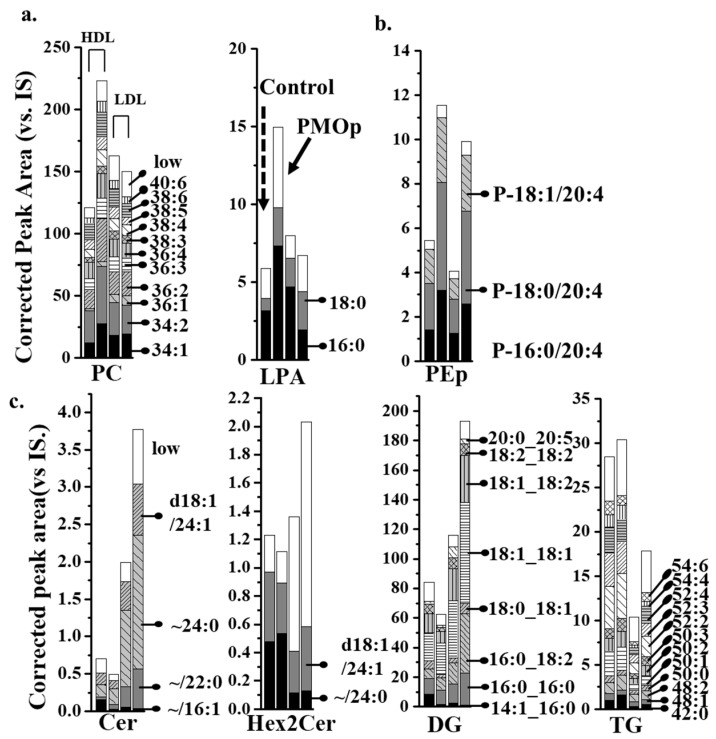
Stacked bar graphs of seven lipid classes showing significant changes (>1.5-fold and *p* < 0.05) in the total amount of lipids between the PMOp and control groups among 19 lipid classes: (**a**) >1.5-fold increase in only HDL of the PMOp group, (**b**) >1.5-fold increase in both HDL and LDL fractions, and (**c**) >1.5-fold increase in only LDL. The lipid species marked with acyl chain structures at the right of each bar are the relatively high-abundance species in each lipid class, and “low” represents the summed amount of the remaining low-abundance species.

**Figure 5 antioxidants-09-00046-f005:**
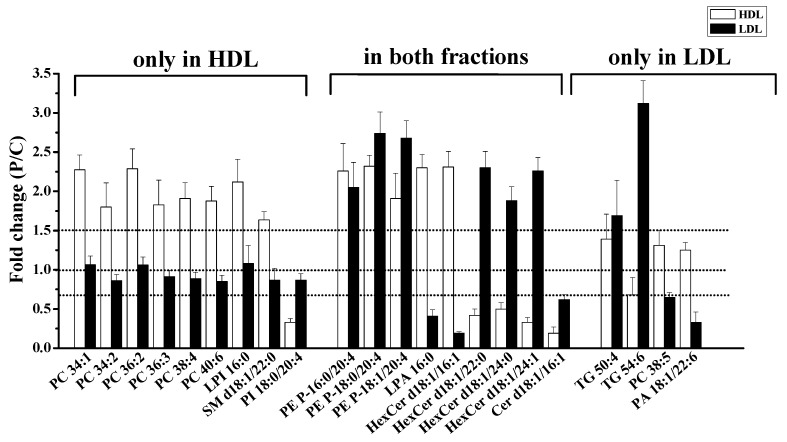
Plot of the fold ratio [PMOp/control (P/C)] of lipid species, showing significant changes (>1.5-fold and *p* < 0.05) in the HDL or LDL fractions, alone or in both fractions.

**Figure 6 antioxidants-09-00046-f006:**
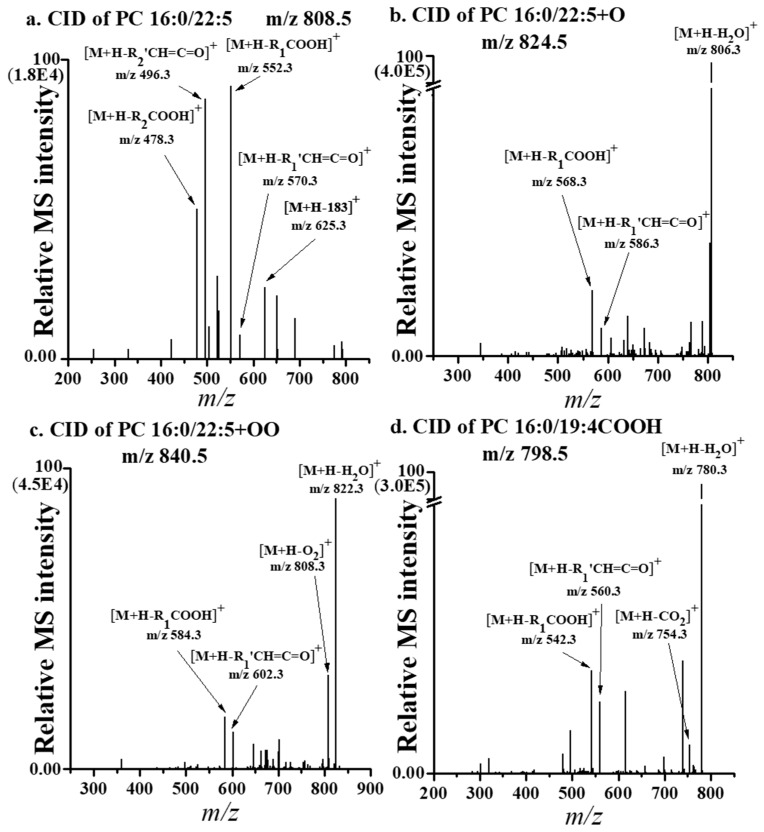
CID spectra of (**a**) intact PC 16:0/22:5 and oxidized products (**b**) PC 16:0/22:5 + O, (**c**) PC 16:0/22:5 + OO, and (**d**) PC 16:0/19:4COOH.

**Table 1 antioxidants-09-00046-t001:** Fold ratio [PMOp/control (P/C)] of oxidized phospholipids identified in the LDL fraction obtained by nUHPLC–ESI–MS/MS. Underlined species represent the chain structure of intact molecules, and species in boldface represent >1.5-fold change. * = *p* < 0.05 and ** = *p* < 0.01.

Class	Chain Structure		*m/z*	Control (*n* = 10)	PMOp (*n* = 10)	P/C
	R1	R2				
**PC**	34:5		752.5	0.02 ± 0.00	0.01 ± 0.00	**0.43 ± 0.05 ***
(20)	14:0	8:1 COOH	622.5	0.01 ± 0.00	0.01 ± 0.00	1.16 ± 0.24 **
	14:0	5:0 COOH	582.5	0.01 ± 0.00	0.02 ± 0.00	1.45 ± 0.37
	38:5		808.5	9.25 ± 0.65	6.20 ± 0.36	**0.65 ± 0.06 ***
	16:0	22:5 + O	824.5	0.02 ± 0.00	0.05 ± 0.01	**2.04 ± 0.38**
		22:5+OO	840.5	0.02 ± 0.00	0.03 ± 0.00	**1.91 ± 0.24 ****
		19:4 CHO	782.5	0.01 ± 0.00	0.01 ± 0.00	1.21 ± 0.43 *
		19:4 COOH	798.5	0.01 ± 0.00	0.02 ± 0.00	**2.41 ± 0.36 ****
		16:3 COOH	758.5	0.02 ± 0.00	0.02 ± 0.00	1.24 ± 0.21
		13:2 COOH	718.5	0.01 ± 0.00	0.02 ± 0.00	1.26 ± 0.37
		10:1 COOH	678.5	0.04 ± 0.00	0.05 ± 0.00	1.34 ± 0.19
		7:0 COOH	638.5	0.01 ± 0.00	0.01 ± 0.00	0.96 ± 0.43 *
	16:1	22:4 + O	824.5	0.02 ± 0.00	0.05 ± 0.00	**1.92 ± 0.30**
		16:2 COOH	758.5	0.03 ± 0.00	0.03 ± 0.00	1.02 ± 0.11 **
	18:0	20:5 + O	824.5	0.02 ± 0.00	0.04 ± 0.01	**1.87 ± 0.36**
		20:5 + OO	840.5	0.02 ± 0.00	0.03 ± 0.00	**1.93 ± 0.31 ****
	18:1	20:4 + O	824.5	0.03 ± 0.00	0.05 ± 0.00	**2.10 ± 0.28 ***
		20:4 + OO	840.5	0.02 ± 0.00	0.03 ± 0.00	**1.85 ± 0.26 ****
		14:2 COOH	758.5	0.02 ± 0.00	0.03 ± 0.00	1.25 ± 0.26 *
	18:2	17:2 COOH	798.5	0.01 ± 0.00	0.02 ± 0.00	**2.16 ± 0.43 ****
		14:1 COOH	758.5	0.02 ± 0.00	0.02 ± 0.00	0.99 ± 0.11
		11:0 COOH	718.5	0.03 ± 0.00	0.04 ± 0.01	1.22 ± 0.26
PA	16:0	18:2	671.5	0.51 ± 0.06	0.17 ± 0.07	**0.34 ± 0.14 ***
(6)	16:0	18:2 + O	687.5	0.05 ± 0.01	0.04 ± 0.00	0.81 ± 0.15
		13:3 CHO	565.5	0.04 ± 0.01	0.04 ± 0.01	1.05 ± 0.27
		10:2 CHO	525.5	0.13 ± 0.02	0.10 ± 0.02	0.82 ± 0.19
	18:1	22:6	745.5	5.29 ± 0.36	1.74 ± 0.67	**0.33 ± 0.13 ***
	18:1	22:6 + OO	777.5	0.63 ± 0.09	0.96 ± 0.06	**1.54 ± 0.25 ***
		19:5 CHO	719.5	0.03 ± 0.00	0.06 ± 0.01	**1.85 ± 0.26 ***
		19:5 COOH	735.5	0.24 ± 0.06	0.24 ± 0.04	0.98 ± 0.28
PI	18:0	18:2	861.5	0.54 ± 0.07	0.36 ± 0.03	**0.66 ± 0.11**
(7)	18:0	18:2 + OO	893.5	0.00 ± 0.00	0.01 ± 0.00	1.12 ± 0.06
		15:1 COOH	851.5	0.00 ± 0.00	0.00 ± 0.00	1.44 ± 0.16
		12:0 COOH	731.5	0.00 ± 0.00	0.01 ± 0.00	**1.74 ± 0.23**
	18:0	22:4	913.5	0.10 ± 0.03	0.07 ± 0.01	**0.66 ± 0.20**
	18:0	16:2 CHO	847.5	0.01 ± 0.00	0.01 ± 0.00	0.91 ± 0.33
		16:2 COOH	863.5	0.04 ± 0.01	0.04 ± 0.00	1.06 ± 0.20*
		15:1 CHO	835.5	0.00 ± 0.00	0.00 ± 0.01	1.10 ± 0.22
		13:1 CHO	807.5	0.02 ± 0.00	0.03 ± 0.00	1.22 ± 0.05*

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
