# Peer review of "Perturbations of Lipids and Oxidized Phospholipids in Lipoproteins of Patients with Postmenopausal Osteoporosis Evaluated by Asymmetrical Flow Field-Flow Fractionation and Nanoflow UHPLC–ESI–MS/MS"

_antioxidants, 2020, doi:10.3390/antiox9010046_

Round 1

Reviewer 1 Report

The study is based on a small sample size, but its conclusions are resonant.

I suggest the authors to specify that participants were free from the use of drugs (e.g. bisphosphonates or statins)  

Author Response

The study is based on a small sample size, but its conclusions are resonant.

I suggest the authors to specify that participants were free from the use of drugs (e.g. bisphosphonates or statins)  

--> Upon the suggestion, a sentence was added at the line 96 as

Participants were free from the use of drugs such as bisphosphonates or statins.”

Reviewer 2 Report

The authors performed a comprehensive lipidomic analyses of lipoproteins separated from plasma of PMOp and control subjects. The study is well designed and the data are very nicely and logically presented and discussed. The major finding of the study is elucidation of the lipidomic fingerprint of  osteoporosis which in future, in combination with appropriate high-speed technology might  be used for diagnosis and estimation of the disease progression.

Minor comments:

1. If data or plasma are available the data shown in table S1 should be extended and laboratory parameters in particular total cholesterol, triglycerides, HDL-c, LDL-c, CRP, IL-6 should be included.

Author Response

Minor comments:

If data or plasma are available the data shown in table S1 should be extended and laboratory parameters in particular total cholesterol, triglycerides, HDL-c, LDL-c, CRP, IL-6 should be included.

-->  Biobank staffs of Chungbuk National University Hospital (Cheongju, Korea) and Chungnam National University Hospital (Daejeon, Korea) confirmed us that plasmas are well selected from subjects without any disease. But we were told that subjects did not undergo profound tests such as measuring cholesterol levels and CRP etc.

Reviewer 3 Report

The study by Dr Lee and colleagues isolated plasma HDL and LDL using AF4 and profiled those lipoproteins’ lipid components with nUHPLC-ESI-MS/MS. The authors first studied the lipoprotein lipid profiles using untargeted shotgun approach then quantified the oxidized lipid species using targeted SRM method.  Finally, the authors identified differential lipid profiles particularly oxidized species in lipoproteins of postmenopausal patients with/without osteoporosis. In general, the manuscript was well written and the findings are interesting, however, it does not seem a complete study even though the authors admitted its preliminary nature. I have the following points to address.

Major concerns

Combining AF4 and LC-MS/MS techniques to evaluate the lipid profiles of LDL/HDL particles has been previously reported, but the prior studies were not referred in this manuscript. The authors may need to address the technology novelty if they intend to present their work as a methodology manuscript. In addition, the isolation of lipoproteins should be validated by ApoA1 vs ApoB100. The LDL and ox-LDL may differ in size, does semi-prep AF4 isolation potentially discriminate these two different particles? Recovery of LDL versus ox-LDL from AF4 isolation needs to be examined, in particular, the oxidation of LDL has been previously reported correlating with osteoporosis. The author may need to examine the purity of isolated lipoproteins against contamination from plasma protein complexes or lipids. Are the increase in lipids from PMOp patients due to increased lipoproteins or due to changes in lipid composition of lipoproteins? The lipid data may be better normalized against lipoprotein particle numbers and respective lipoprotein markers (ApoA1 and ApoB100). The study identified interesting association of lipoprotein lipidomics with postmenopausal osteoporosis without any casual experimental data provided. However, the data seems over extrapolated. For example, on page 8, “PEp is known to protect the unsaturated acyl chain of lipids against oxidative damage from ROS [31].” How did the authors conclude that “Up- regulation of these PEp species is associated with increased oxidative damage from ROS during the development of osteoporosis to cope with OS.”? I would conclude the opposite. Page 6. The anti-inflammatory lipid PI 18:0/20:4 was decreased in HDL in PMOp samples, the authors stated “A decrease in the most abundant PI 18:0/20:4 in HDL can be thought of as an increased conversion of PI species to arachidonic acid in patients with PMOp.” Can the authors measure arachidonic acid in the plasma or lipoproteins to support their speculation?

Minor points

The authors used 5 μg/μL extracted lipids for analysis, however, the total plasma LDL level is two-fold higher in PMOp samples. Was the difference in total lipids taken into account for data analysis? Page 2, “Lipid molecules whose levels were significantly different between patients and healthy controls were determined to be the lipoprotein- specific lipids that are relevant to PMOp.” But in Results and Figures, all data were relevant to control group. Page 6, “PI plays an anti-inflammatory role by regulating the immune response by delivering arachidonic acid (20:4), a precursor of eicosanoids that are inflammatory mediators [30].” is difficult to follow. Any means were taken to prevent lipid oxidation during sample processing? Did the authors calculate sample size? 10 subjects per group for human sample studies are risky to get reliable conclusion.

Author Response

I appreciate the fruitful comments and suggestions made by the reviewers. In the following, I have written my replies to the reviewers’ comments in blue and newly added or edited sentences in red and unchanged (original) sentences in black. Line, figure, and table number in the following responses are based on the revised manuscript.

Major concerns

Combining AF4 and LC-MS/MS techniques to evaluate the lipid profiles of LDL/HDL particles has been previously reported, but the prior studies were not referred in this manuscript. The authors may need to address the technology novelty if they intend to present their work as a methodology manuscript.

--> Upon suggestions, references were individually marked with the addition of the following phrase “as reported by the previous studies” at the line 62 as

“HDL and LDL particles from the plasma of patients with PMOp were size-sorted by semi-preparative scale asymmetrical flow field-flow fractionation (asymmetrical FlFFF or AF4), an elution-based method which separates macromolecules or particulate species by size [20-24]. Then, collected fractions of narrow size distribution can be used for proteomic or lipidomic analysis as reported by the previous studies [24,25].

In addition, the isolation of lipoproteins should be validated by ApoA1 vs ApoB100.

--> Validation with ApoA1 and ApoB100 can be helpful for confirmation, but the verification of HDL and LDL lipoprotein fractions was done by comparing retention time with HDL and LDL standards. For the confirmation of lipoproteins during AF4 separation, the human plasma samples were stained with Sudan Black B prior to AF4 separation and detected at 600 nm at which wavelength proteins are not detected and only stained lipoproteins can be detected. However, when collecting lipoprotein particles for lipid analysis, plasma samples without staining were injected to AF4. This was added in the section 3.1.

(line 175-177) “For the confirmation of lipoprotein separation, plasma samples were stained with Sudan Black B prior to AF4 separation and detected at 610 nm.”

(line 186-187) “When lipoprotein fractions were collected, plasma samples without staining with SBB were injected to AF4.”

The LDL and ox-LDL may differ in size, does semi-prep AF4 isolation potentially discriminate these two different particles? Recovery of LDL versus ox-LDL from AF4 isolation needs to be examined.

--> Size difference between LDL and ox-LDL is not large enough to be resolved by AF4 due to the limitation in resolution. Therefore, the recovery of LDL vs. Ox-LDL could not be distinguished.

in particular, the oxidation of LDL has been previously reported correlating with osteoporosis. The author may need to examine the purity of isolated lipoproteins against contamination from plasma protein complexes or lipids.

--> Plasma samples were treated with a ProteoPrep® Immunoaffinity Albumin & IgG Depletion Kit to remove high abundance IgG and albumin first before they were injected to AF4. Remaining proteins in the lipoprotein fraction including the proteins associating with lipoprotein particles were expected to be removed during the Folch-based lipid extraction procedures including the centrifugation method with Amicon Centrifugal Filters.

Are the increase in lipids from PMOp patients due to increased lipoproteins or due to changes in lipid composition of lipoproteins? The lipid data may be better normalized against lipoprotein particle numbers and respective lipoprotein markers (ApoA1 and ApoB100).

--> The increase in lipids from PMOp patients was due to the compositional change since lipid amount was relatively compared with the basis of a same amount of total lipids in each lipoprotein between the patients and controls. After extraction of lipids in each lipoprotein, lipids were dried and measured. Then lipid concentration was adjusted at 5 μg/μL prior to analysis. This was explained in the experimental section 2.4.

(line 124) “After lipid extraction, lipids in organic solvents were transferred to a 2 mL tube and sealed with 0.45 μm MilliWrap PTFE membrane from Millipore to avoid lipid evaporation during lyophilization for 12 h. Then, dried lipid powders (1.0~1.2 mg) were reconstituted in CH3OH:CHCl3 (9:1, v/v) to prepare a stock solution for each fraction which was stored in a -80 °C freezer. The stock solution was diluted with CH3OH:H2O (8:2, v/v) at a final lipid concentration of 5 μg/μL prior to analysis.”

The study identified interesting association of lipoprotein lipidomics with postmenopausal osteoporosis without any casual experimental data provided. However, the data seems over extrapolated. For example, on page 8, “PEp is known to protect the unsaturated acyl chain of lipids against oxidative damage from ROS [31].” How did the authors conclude that “Up- regulation of these PEp species is associated with increased oxidative damage from ROS during the development of osteoporosis to cope with OS.”? I would conclude the opposite.

--> It is not clearly understood why the increase of PEp was associated with the menopausal women with low bone mineral density. It may be expected that the increase in the oxidative damage causes the upregulation of PEp. Because the sentence can be rather absurd, it was removed by rewriting into (line 255-262) as

-->  “The increase in PEp was similarly reported in menopausal women with low bone mineral density (BMD) compared to the BMD in women with normal bone density in an earlier study [12]. While the last study showed a significant increase of PE P-18:0/20:4 species, the present work showed a more detailed information on the accumulated PEp species.”

Page 6. The anti-inflammatory lipid PI 18:0/20:4 was decreased in HDL in PMOp samples, the authors stated “A decrease in the most abundant PI 18:0/20:4 in HDL can be thought of as an increased conversion of PI species to arachidonic acid in patients with PMOp.” Can the authors measure arachidonic acid in the plasma or lipoproteins to support their speculation?

 --> There are certain methods to measure the quantity of arachidonic acid, however our study was focused on analyzing lipids without including free fatty acids since simultaneous analysis of free fatty acids with other glycerophospholipids and glycerolipids is complicated due to the similarity in retention times of fatty acids and lysophospholipids, and the relatively poor ionization efficiency of fatty acids under the present run conditions that were specialized for the analysis of glycerophospholipids and glycerolipids. The relevant sentences in the line 236-239 on arachidonic acid was removed.

Minor points

The authors used 5 μg/μL extracted lipids for analysis, however, the total plasma LDL level is two-fold higher in PMOp samples. Was the difference in total lipids taken into account for data analysis? Page 2, “Lipid molecules whose levels were significantly different between patients and healthy controls were determined to be the lipoprotein- specific lipids that are relevant to PMOp.” But in Results and Figures, all data were relevant to control group.

--> As reviewer pointed, the peak area of LDL in the fractogram was larger in PMOp samples than controls. But after the extraction process, we adjusted the lipid concentration of all fractions (HDL and LDL) to 5 μg/μL for nUHPLC-ESI-MS/MS analysis, not normalized to the lipoprotein level.

Page 6, “PI plays an anti-inflammatory role by regulating the immune response by delivering arachidonic acid (20:4), a precursor of eicosanoids that are inflammatory mediators [30].” is difficult to follow.

--> Same as the answer to the question 7 at the above.

Any means were taken to prevent lipid oxidation during sample processing? Did the authors calculate sample size?

EP tube was covered with 0.45 μm MilliWrap PTFE membrane to minimize oxidation and lipid evaporation during lipid lyophilization after lipid extraction. Then dried lipids were weighed. This was added to the line 125 as

“After lipid extraction, lipids in organic solvents were transferred to a 2 mL tube and sealed with 0.45 μm MilliWrap PTFE membrane from Millipore to avoid lipid evaporation during lyophilization for 12 h. Then, dried lipid powders (1.0~1.2 mg) were reconstituted in CH3OH:CHCl3 (9:1, v/v) to prepare a stock solution for each fraction which was stored in a -80 °C freezer. The stock solution was diluted with CH3OH:H2O (8:2, v/v) at a final lipid concentration of 5 μg/μL prior to analysis.”

10 subjects per group for human sample studies are risky to get reliable conclusion.

--> As reviewer commented, the sample numbers were small. We aimed to carry out a pilot study with a small number of samples without collecting HDL and LDL fractions of a large number of samples which may take a long period of time. In order to validate the usefulness of the candidate lipids, a further systematic investigation with a large cohort will be needed as we mentioned in the line 381-384.

Round 2

Reviewer 3 Report

The authors have addressed most of my concerns and improved the data interpretation. I have no further comments.